# Crack-Resistant Cements under Drying: Results from Ring Shrinkage Tests and Multi-Physical Modeling

**DOI:** 10.3390/ma15124040

**Published:** 2022-06-07

**Authors:** Vít Šmilauer, Pavel Reiterman, Rostislav Šulc, Petr Schořík

**Affiliations:** 1Department of Mechanics, Faculty of Civil Engineering, Czech Technical University in Prague, Thákurova 7, 166 29 Prague , Czech Republic; 2Experimental Center, Faculty of Civil Engineering, Czech Technical University in Prague, Thákurova 7, 166 29 Prague, Czech Republic; pavel.reiterman@fsv.cvut.cz; 3Department of Construction Technology, Faculty of Civil Engineering, Czech Technical University in Prague, Thákurova 7, 166 29 Prague, Czech Republic; rostislav.sulc@fsv.cvut.cz; 4Českomoravský Cement, a.s., Mokrá 359, 664 04 Mokrá-Horákov, Czech Republic; petr.schorik@cmcem.cz

**Keywords:** crack-resistant cements, ring shrinkage test, chemical shrinkage, drying shrinkage, creep, strength gain, cracking, hygro-mechanical simulation

## Abstract

Cementitious materials exhibit shrinkage strain on drying, leading easily to crack formation when internally or externally restrained. It is known that cements with a slow strength gain show higher crack resistance under external drying. The ring shrinkage test can be considered an accelerated method for cracking tendency due to existing historical correlations between ring cracking time and long-term surface concrete cracking. The experimental campaign used ring shrinkage tests on 25 mortars, covering 10 commercial cements and 15 cements produced on demand, covering Portland cements and blended cements up to a 30% slag substitution. The results show that the restrained ring cracking time generally increases with lower Blaine fineness and higher slag substitution in 6 to over 207 days’ span. Upper limits for crack-resistant cements were proposed for 2-day compressive strength and Blaine fineness, in the case of Portland cements: 27.7 MPa and 290 m^2^/kg, respectively. A hygro-mechanical model successfully replicated strain evolution with crack formation and brittle failure. Only two out of ten commercial cements were classified as crack-resistant, while the ratio increased to 10 out of 15 cements which were produced on demand.

## 1. Introduction

The volume shrinkage of concrete presents an important driving force for potential crack formation due to internal or external restraints. Chemical shrinkage presents a well-known phenomenon, where the *C*_3_*S* hydration itself leads to a volume decrease by approximately 9% [1]. The second, diffusion-driven phenomenon covers drying shrinkage quantified by the shrinkage strain. Experiments on 4 × 8 × 32 mm and 50 × 50 × 400 mm prisms led to linear shrinkage strains over 3000 × 10^−6^, 1400 × 10^−6^ and 500 × 10^−6^ at 500 days for the cement paste, mortar and concrete, respectively; the samples had *w/c* = 0.50 and were exposed to 48% of relative humidity [2]. The relative humidity and the *w/c* ratio were found as the two dominant factors controlling the magnitude of drying shrinkage [3,4].

Aging creep acts beneficially against crack formation due to stress relaxation. This was already recognized by G. Pickett in 1942, who pointed that, under most conditions, concrete would have severe cracking if there were no creep [5]. Another beneficial proof of creep is the presence of surface cracking and the absence of deep cracks under drying, when more time for stress relaxation occurs towards a sample core [6]. Creep can be increased by slowing down the hydration, as demonstrated by a fictitious degree of hydration method [7]. Slow hydration gives rise to crack-resistant cements under drying environments, as witnessed by several authors [8,9,10].

Systematic studies on durability carried out in the USA after the 1930s stated that concrete cracking is the most predominant factor, resulting from the use of high early strength cements [10]. The need for fast construction schedules generally increased the C_3_S contents and raised the Blaine fineness in cement while decreasing the water-to-cement ratio (*w/c*); such changes led from crack-resistant to crack-prone concretes [8]. The shift can be manifested in the compressive strength evolution of Type I general purpose Portland cements in the US between the 1950s and 1994, see Figure 1 [11]. Although the strength data used different historical versions of ASTM C109, the ratio of 1:28 days has obviously increased. The study of the Iowa highway deterioration led John Lemish to conclude in 1969 that “Concretes that gain strength slowly are related to good performance” [8].

Several recommendations have been made for crack-resistant cements when exposed to drying. For example, slow-hardening cements were preferred for German concrete pavements in 1936 [12]. P. H. Bates designed Type II cement within ASTM C 150 in 1940, limiting 7-day strength to approximately 15 MPa (2200 psi) [9]. R. Breitenbucher developed a restrained frame, later standardized in RILEM TC 119 TCE [13]. He found relevant factors increasing the cracking resistance of concrete: low fresh concrete temperature, low cement fineness, a higher *w/c* ratio, low alkalies and low C_3_A content. Based on that research, three criteria were proposed for crack-resistant cement: concrete compressive strength below 7 MPa at 12 h of semi-adiabatic curing, chemical shrinkage below 1.05 mL per 100 g of cement at 12 h and visual inspection of dried cement pastes [14].

### 1.1. Surface Cracking

The problems arising from high early strength gain cements can be demonstrated on the surface cracking of concrete pavements. Figure 2 shows the statistics from approximately a half of all Czech concrete pavements by the laser crack measurement system (LCMS). The scanned pavements were cast mostly after 1990, and LCMS covers 887 lane-km out of ∼1650 current lane-km [15]. A total of 75% of such concrete pavements show visible cracking after 15 years, decreasing the service life from 40–50 to 25–30 years. Several sections were removed even after a few years due to surface cracking. This is in contrast with old concrete pavements, particularly the D1 highway built in the 1970s and 1980s, where the concrete has lasted well over 40 years without significant cracking problems. The major difference lies in increased cement reactivity, showing similar trends as in Figure 1. The old cement reached flexular strength 4.8 MPa at 3 days while the same strength today is reached at 1.5 day [15].

Figure 3 shows two examples of concrete surface cracking: a plain concrete pavement on the D1 highway km 237.400 cast in 2004 and a reinforced parapet wall on Prague Dejvice campus built in 2008, both in the Czech Republic. Visible cracking due to drying appears generally weeks to months after drying [16], and, in many cases, even after years; see Figure 2. High speed construction schedules resulting in the elimination of slowly hydrating cements lead to omnipresent surface cracking, impairing the durability and service life, especially for exterior concrete elements [9,10].

### 1.2. Microtomography

The recent 3D X-ray microtomography experiments from Imperial College London proved that invisible microcracks originate during sealed hydration, detected already at 1 day [17,18,19]. The mean microcrack widths were found as 10–20 μm after 3.6 years of sealed hydration: the crack density and connectivity increased with the decreasing *w/c* ratio and with the increasing cement reactivity. Despite more microcracking at lower *w/c* ratios, the oxygen permeability decreased, which shows that transport occurred primarily through the paste and not the microcracks.

The microcracks formed during sealed hydration tend to coalescence into visible cracks when concrete is exposed to drying. The crack growth stability condition postulates that every *n*th crack needs to close, causing the opening of other cracks [20]. The same mechanism was described by a holistic model of concrete deterioration by P. K. Mehta [21], showing a transition from discontinuous microcracks to interconnected microcracks, accompanied with concrete spalling and disintegration when exposed to cyclic environmental actions.

### 1.3. Ring Tests

The restrained ring test is a well-established method for testing the crack resistance of cements, mortars or concretes, used since 1939 by R. Carlson [8]. The test was documented historically on at least 13 different ring dimensions and became adopted in several standards such as ASTM C1581 or AASHTO T334 [22]. The test reflects the tensile stresses induced by restrained autogenous and drying shrinkage. Stress relaxation due to creep acts beneficially for crack mitigation. The mortars are more sensitive to restrained shrinkage than concrete, generally due to a higher cement content and higher permeability. The results from mortars are also indicative for concrete, since it is the same cement paste responsible for autogenous and drying shrinkage [23].

The ring shrinkage test provides quantitative cracking time of a cementitious system on a particular geometry, which is useful for mutual comparison and optimization [22]. The limitations of several ring setups are in sealing, which might be imperfect and cause additional drying [24]. The results from cement paste need to be scaled for mortar or concrete since the paste has lower permeability, higher creep and shrinkage [8]. Drying starts usually after one day of sealed curing which causes a different degree of hydration due to different hydration kinetics. Repeatability of the ring test can be a critical point. It was explained to some extent by material stochastic nature [25].

The cracking time in the restrained ring test has a strong correlation with long-term concrete cracking and performance [8]. Such an experiment was conducted on 27 various cements used for 104 panels, 2.74 × 1.22 × 0.41 m in size, placed around the Green Mountain Dam, Colorado, in 1943. The cracking time of the mortar ring correlated well with the concrete surface cracking after 53 years; the sooner the ring cracked, the more severe the cracking appeared later. Low alkali cements, coarser cements and lower C_3_A cement performed the best of all. Similar findings were supported by testing fine and coarse cement in dual-ring paste measurement, where finer cement cracked at 87 h while the coarse one showed 6× lower tensile stress and did not crack to the end of the test [26]. The ring shrinkage test presents a unique short-term method how to access long-term cracking tendency of concrete structures subjected to drying. The test combines drying shrinkage, aging creep, evolution of tensile strength and fracture energy, mimicking drying surface area of mortar or concrete [27].

The objective of this paper aims at crack resistance assessment of 25 cements, showing the current situation on the market and possible ways towards crack-resistant cements. For that purpose, 10 existing commercial cements were tested, supplemented with 15 cements ground or blended on demand. The ring test uses mainly mortars 3:1 (sand:cement) with *w/c* = 0.45, containing 510–530 kg/m^3^ of the cement, preventing the segregation of cement grains. To our best knowledge, there are no published data testing dozens of different cements in the last decades consistently on the same ring geometry. This paper closes the gap by exploring crack resistance of 25 cements and demonstrating that it is a matter of preferences rather than cement plant production processes or cost.

## 2. Materials and Methods

The experimental program covers 25 cements from 8 cement plants that were tested between the years 2017 and 2021. Ten cements cover standard products available on the market, and fifteen cements were produced on demand, mainly by decreasing the grinding time leading to lower fineness. The program incorporated Portland-slag blended cement up to 30% substitution to demonstrate the beneficial role of slower slag hydration. The Blaine fineness covers a wide span from 250 to 433 m^2^/kg. The following products from cement plants were involved:15 cements from Mokrá, Czech Republic (4× CEM I, 5× CEM II/A-S, 6× CEM II/B-S);3 cements from Ladce, Slovak Republic (CEM I);1 on-site blended cement Mokrá + slag SMŠ 400 (CEM II/B-S);1 cement from Praha-Radotín, Czech Republic (CEM II/B-S);1 cement from Hranice, Czech Republic (CEM II/B-S);1 cement from Prachovice, Czech Republic (CEM II/B-S);1 cement from Ożarów, Poland (CEM I);1 cement from Rohožník, Slovak Republic (CEM I);1 cement from Kiralyegyháza, Hungary (CEM II/B-S).

The oxide and mineral composition of known selected cements and slags are summarized in Table 1. The mineral composition of CEM I is calculated according to the Bogue standard procedure. The same clinker from a particular cement plant is used in slag-blended cements CEM II/A-S and CEM II/B-S. The chemical composition of blast furnace slag, added to the clinker, is generally unknown to us. The only exception is the Czech ground-granulated blast-furnace slag (GGBFS) SMŠ 400 (Kotouč Štramberk, spol. s r.o., Czech Republic) [28], which was intermixed directly with CEM I 42.5 R (sc) Mokrá.

The workflow of experiments is graphically summarized in Figure 4. Isothermal calorimetry was used on cement pastes, while compressive strength and ring shrinkage tests were operated on mortars.

Isothermal calorimetry was conducted in the TamAIR (Thermometric AB, Stochholm, Sweden) calorimeter at 20 °C. The pastes were mixed externally by hand for approximately 30 s and vibrated in the IKA Vortex I orbital shaker for 20 s. The procedure followed the prEN 196-11 Method “A” external mixing with two modifications. Fourteen pastes were tested according to prEN 196-11 at *w/c* = 0.40, while eleven pastes had a higher *w/c* = 0.45 matching mostly the ring’s mortars. The second modification used heat flow integration from 45 min after mixing. The heat released before 45 min was calculated from the known initial temperature, the estimated heat capacity and the heat up to 45 min. The released heat between 0 and 45 min reached 5–19 J/g of cement. The calorimetry always used two samples with approximately 18 g of cement in each ampoule, and the differences were negligible. The evolution of compressive strength was carried out according to the EN 196-1 standard on three samples.

The dimensions for the ring shrinkage test originate from R. W. Carlson’s design, but it has a thinner steel ring to increase the deformation [29]; see Figure 5. Standard mortars were prepared from cements mostly at *w/c* = 0.45 with a sand:cement ratio of 3:1. Drying at a relative humidity of 45–55% and 19–22 °C started after 24 h of sealed curing. Four strain gauges HBM 1-LY11-10/120 measured the contraction of the steel ring, recorded on a DataTaker’s DT80G datalogger (Thermo Fisher Scientific Australia, Melbourne, Australia) every 30 min. Each cement was tested in at least two rings, and average cracking time and standard deviation were reported. A very thin grease layer between mortar and steel ensured low friction and, in the majority of cases, led to a brittle ring failure without a noticeable softening part.

A hygro-mechanical model proceeds as a staggered problem, solving at first the moisture transport followed by a mechanical model in each time step. The implementation uses a linear viscoelastic creep model B3 combined with an anisotropic fixed-crack model, implemented in the OOFEM software (version 2.5, an in-house software developed at Czech Technical University in Prague) [30,31].

## 3. Results and Discussion

### 3.1. Isothermal Calorimetry

The results from isothermal calorimetry are summarized in Figure 6 and are consistent with published data elsewhere and data from cement manufacturers [1,32]. An asterisk signalizes a crack-resistant cement according to the 40-day criterion of the ring shrinkage test as described in Section 3.2. The differences in released heat are mainly remarkable up to 24 h of hydration. At that time, crack-resistant cements release less heat than 156, 142 and 154 J/g for groups in CEM I, CEM II/A-S and CEM II/B-S, respectively. However, several cements releasing less hydration heat were found to be crack-prone. Note that released heat is proportional to chemical shrinkage, lower for a coarser cement [33].

Apparently, other factors than released heat contribute to crack resistance, e.g., clinker reactivity, cement particle distribution, pore size distribution, extensibility of hydrates or alkali content. In slag-blended systems, slag is generally harder for grinding. The majority of studied cement plants grind clinker and slag together, yielding coarsely ground slag and finely ground clinker, negatively contributing to fast initial chemical shrinkage and reducing the ring cracking time.

### 3.2. Ring Shrinkage Test

The majority of ring tests failed in a brittle manner with a single crack across the whole ring. Figure 7 shows such a characteristic strain evolution from three selected cements. Brittle fracture is evident in the CEM I 42.5 R(sc) cement at 30 days, while a coarsely ground cement to 256 m^2^/kg and a blended cement showed no cracking. The strain fluctuations reflect the RH environment kept in the range of 45–55%.

The results from 25 tested cements are graphically summarized in Figure 8. The description around a data point expresses the Blaine fineness and the average cracking time from at least two rings; the symbol ≥ means that the experiment was terminated with at least one uncracked ring, providing the minimum average cracking time. The red cross marks crack-prone cements that crack under 40 days; this criterion was selected based on CEM I 42.5R(sc) Mokrá from 03/2018, responsible for 75% visible pavement cracking up to 15 years; see Figure 2. The green dots show crack-resistant cements, when the ring cracks after 40 days.

The results are consistent with similar older experiments which were designed to reveal the differences among cements. In 1940, 27 concretes made from 27 various cements gave cracking time in the range between 4 and 20 days on a highly internally restrained ring, using a polished steel disc instead of a ring [8]. Increasing alkali content, cement fineness and C_3_A shortened the cracking time. The same conclusions were found in a cracking frame test [13]. Other papers mutually compared only a few concretes or cements, focusing on mix design, fibers and admixtures effects rather than cement properties or kinetics [34,35,36].

Figure 8 justifies two factors acting beneficially for crack-resistant cements: a low Blaine fineness and clinker substitution by less reactive slag. The violet line is a hypothetical threshold for crack-resistant cements that can withstand at least 40 days in the ring shrinkage test
(1)$$\begin{array}{c} \mathrm{Blaine}\;\mathrm{fineness}\;\left(m^{2}/\mathrm{kg}\right)<290+2.57\times \mathrm{SSL}\left(\%\right), \end{array}$$
where SSL stands for the slag substitution level (limited to 30% by tests and extrapolated).

Equation (Equation 1) implies the maximum fineness of 290, 340 and 380 m^2^/kg for CEM I, CEM II/A-S and CEM II/B-S crack-resistant cements, respectively. Only two crack-resistant commercially produced cements from Figure 8 exist in the Czech market: CEM II/B-S 32.5 R Radotín (Blaine 326 m^2^/kg) and CEM II/B-S 32.5 R Prachovice (Blaine 343 m^2^/kg). A particular batch of CEM I 42.5 R(sc) Mokrá (Blaine 312 m^2^/kg) from 11/2019 also showed high crack resistivity; however, this is an exception to the other tested batches from the same plant. A careful analysis of the clinker composition, strength gain and alkalies did not reveal any remarkable difference against long-term production, and there is no explanation for this particular batch.

The rest of the green-colored cements could be produced easily by modifying standard industrial processes and at almost the same cost. Those crack-resistant cements disappeared from the market due to the requirements of fast construction schedules and a general unawareness and ignorance of long-term cracking problems and their relationship with the strength gain.

### 3.3. Compressive Strength

Table 2 summarizes the Blaine fineness, compressive strength at 2 and 28 days, released heat at 24 h and the average ring crack time with standard deviation. Crack-resistant cements show a 2-day compressive strength less than 27.7, 22.5 and 20.1 MPa for groups in CEM I, CEM II/A-S and CEM II/B-S, respectively. However, there are several cements satisfying those criteria and cracking under 40 days. The strength correlates well with the Blaine fineness in each group of CEM I, CEM II/A-S and CEM II/B-S, reflecting also different clinker reactivity from different cement plants.

### 3.4. Hygro-Mechanical Model

A hygro-mechanical model is able to capture the ring behavior, as demonstrated in the COST TU1404 benchmark [27]. The relative humidity field can be obtained from the water mass balance equation
(2)$$\frac{\partial w}{\partial h}\frac{\partial h}{\partial t}=\nabla \cdot \left[c(h)\nabla h\right],$$
where *w* is the water mass per unit volume, *h* is the relative humidity field and *c(h)* is the moisture permeability function. The simulation for the CEM I 42.5 R(sc) Mokrá cement used the Bažant–Najjar moisture permeability function [37], calibrated as
(3)$$c\left(h\right)=2.56\times 10^{-3}\left(0.1+\frac{1-0.1}{1+{\left(\frac{1-h}{1-0.7}\right)}^{10}}\right)\mathrm{kg}/m/\mathrm{day}.$$

For simplicity, the desorption isotherm used a constant slope $\frac{\partial w}{\partial h}=196$ kg/m${}^{3}$, the initial condition assumed a relative humidity of 0.98, the surface flux used the Newton boundary condition with ambient relative humidity of 0.50 and a hygric exchange coefficient of $h_{w}=0.28$ kg/m${}^{2}$/day.

The solution of the mechanical problem uses a staggered approach with a known relative humidity field at a particular time step. A fixed crack model is combined with a viscoelastic model to obtain the stress evolution and fracturing strain as
(4)$$\sigma_{i+1}=\sigma_{i}+{\bar{D}}_{ve}\left(\Delta \varepsilon -\Delta \varepsilon^{\prime \prime }-\Delta \varepsilon_{sh}-\Delta \varepsilon_{T}-\Delta \varepsilon_{cr}\right),$$
where ${\bar{D}}_{ve}$ is the incremental tangent stiffness matrix, $\Delta \varepsilon^{\prime \prime }$ is an inelastic strain increment vector, $\Delta \varepsilon_{sh}$ is the autogenous and shrinkage strain increment and $\Delta \varepsilon_{cr}$ is the cracking strain increment [30,31].

The viscoelastic model is based on the B3 solidification model, extended further by the microprestress theory [38,39]. Such an approach allows defining the constitutive law in a material point instead of the average behavior over the cross-section. Figure 9 shows schematically the Kelvin unit chain where the relative humidity controls viscosity and the flow term $\varepsilon_{f}$. In addition, the decrease of relative humidity slows down the equivalent time, which captures the creep reduction of dry concrete. Extensive calibration of the B3 model on the Northwestern University creep and shrinkage database shows that basic and drying creep are inversely proportional to compressive strength [40]. Therefore, concretes with slowly hydrating cements will achieve lower 28-day strength and higher creep.

The rate of the drying shrinkage strain is related to the rate of relative humidity and was calibrated as
(5)$$\begin{array}{c} {\dot{\varepsilon }}_{sh}=k_{sh}\dot{h}=1.42\times 10^{-3}\dot{h}, \end{array}$$
where $k_{sh}$ is the shrinkage coefficient. Autogenous shrinkage is neglected and drying shrinkage strain is the only load in the mechanical problem. The uniaxial tensile strength at 28 days is estimated as 4.0 MPa, the fracture energy at 28 days is estimated as 50 J/m${}^{2}$. The B3 model uses standard values from the mix design for creep behavior [39].

The finite element analysis of a restrained ring approximates geometry with a quarter of the top-half symmetric part. Both hygro-mechanical tasks use the same mesh, containing 2880 hexahedral elements with linear interpolation functions; see Figure 10. First, 2D mesh was created on the cross-section with finer elements towards the surface. Second, 2D mesh rotation by 6${}^{\circ }$ steps created a quarter of the top-half symmetric part. The moisture transport problem defines zero flux on the planes of symmetry, and drying occurs from circumference, top (and symmetrical bottom) according to the experimental setup; see Figure 5a.

Interface elements are placed between the steel and mortar rings, allowing the interface opening and a more compliant shear slip. Mechanical boundary conditions prevent normal displacements on the planes of symmetry. In order to increase the strain localization, an artificial notch was created at the right symmetry plane by reducing the cross-section area by 5% from the exterior circumference. If not done so, damage appears gradually from the exterior surfaces, creating an unrealistic plateau in the steel hoop strain and leading to gradual softening with a much later brittle failure.

Figure 11 shows the relative humidity field and the first principal stress in the mortar ring. In the experiment, two rings cracked at the same time of 30 days. Figure 11a covers the situation at 30 days, just prior to the macrocrack formation over the mortar ring. Figure 11b displays the cracked ring at 31 days, with a noticeable stress drop due to the macrocrack opening and the distorted circular shape.

Figure 12a validates the strain evolution and the brittle ring failure at 30 days. The ring had started drying after 1 day of hydration, which induced hoop strain on the steel ring. Microcracking occurs at exterior drying surfaces. At 30 days, the mortar ring carried the maximum load and further drying led to the formation of a macrocrack and brittle failure. The simulation still retains a small hoop strain in the steel ring due to a nonzero shear stress between the steel and the mortar caused by a small shear stiffness of the contact elements.

The evolution of the hoop stress on the mid-plane testifies that circumferential drying leads to surface softening and microcracking as soon as at 3 days; see Figure 12b. The tensile stress progresses to the interior parts as the drying front moves inside the ring. In addition, Figure 12b provides tensile strength evolution at the interior parts, where the equivalent time reaches the highest values due to the highest relative humidity.

The presented 1/8 model is convenient due to the boundary conditions in orthogonal directions, a small number of degrees of freedom and the computation speed. However, the elastic energy release into a single macrocrack is underestimated since the current model assumes a second symmetric macrocrack formed on the opposite side. Due to the brittle nature of the fracture, the softening part plays no role in this case and the cracking time is predicted correctly.

The limitations of the presented hygro-mechanical model are in estimated parameters which would need to be identified experimentally. They include drying kinetics, estimated for example by the ring’s mass loss, moisture permeability function, the shrinkage coefficient, aging creep, evolution of tensile strength and fracture energy. Successful simulations of the ring test were demonstrated previously with experimentally calibrated parameters [27].

## 4. Conclusions

This paper quantified crack resistivity of 25 cements using the ring shrinkage test with external drying. The 40-day threshold for a crack-resistant cement was deduced from surface cracking of Czech highway concrete pavements. Cements with slower hydration, i.e., with lower Blaine fineness or higher slag substitution level, were generally found to be more crack-resistant on drying than cements with a high early strength gain. Such finding is consistent with the literature [8,9,10,13,26].

Isothermal calorimetry provides only a rough estimation for the cracking tendency of a cement. The most relevant information is the released heat at 24 h, where the difference among cements is the most remarkable. Crack-resistant cements release less heat than 156, 142 and 154 J/g for groups in CEM I, CEM II/A-S and CEM II/B-S, respectively. Similar findings exist for 2-day strength, which needs to stay under 27.7, 22.5 and 20.1 MPa for groups in CEM I, CEM II/A-S and CEM II/B-S MPa, respectively.

The Blaine fineness as a function of the slag-substitution level provides a good measure for crack-resistant cements, summarized in Equation (Equation 1). In that case, the fineness needs to remain below 290, 340 and 380 m^2^/kg for CEM I, CEM II/A-S and CEM II/B-S cements, respectively.

Only two out of ten commercial cements were classified as crack-resistant, both belonging to the CEM II/B-S 32.5 R class. It is no surprise that fast construction schedules and a general unawareness and ignorance of long-term cracking problems eliminated crack-resistant cements from the market. The tests show that crack-resistant cements could be easily produced on demand as 32.5 R and 42.5 N classes, which is proved in 10 out of 15 cements produced by standard cement manufacturing processes. Further research may include detailed experimental identification of material properties, microcrack characterization and crack resistance of blended cements.

The deterioration of several concrete structures exposed to drying is accelerated by cracking. Relevant tests revealing cracking susceptibility, such as the ring shrinkage test, need to be selected to prove durable or sustainable concrete structures. Strength-based criteria, commonly used in the last decades, should become one part of a holistic approach for assessing the long-term performance [10].

## Figures and Tables

**Figure 1 materials-15-04040-f001:**
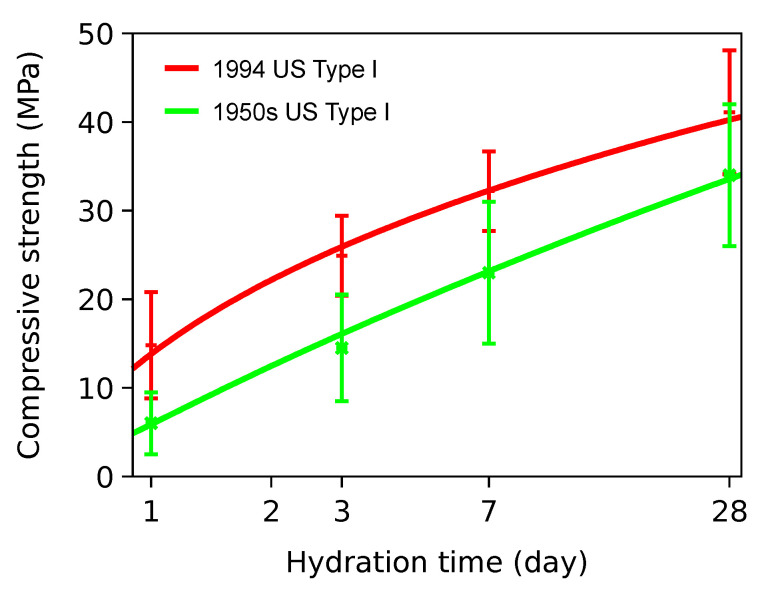
Compressive strength of cements produced in the 1950s (193 cements) and 1994 (>2150 cements) [11].

**Figure 2 materials-15-04040-f002:**
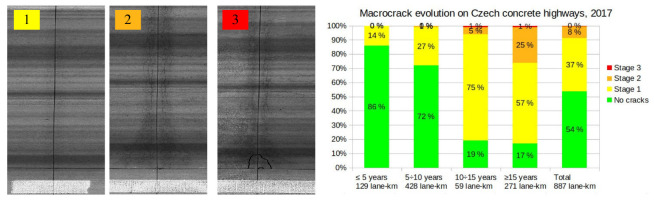
Surface cracking of Czech concrete pavements built mostly after 1990 [15].

**Figure 3 materials-15-04040-f003:**
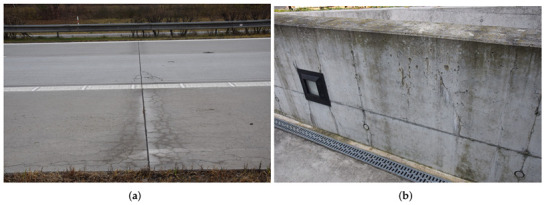
Cracks in a concrete pavement after 18 years (**a**) and in a parapet wall after 14 years (**b**).

**Figure 4 materials-15-04040-f004:**
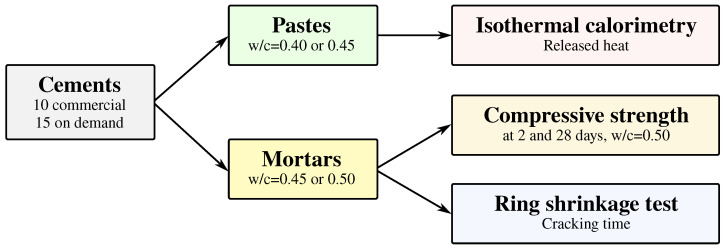
Workflow of experiments for testing 25 cements.

**Figure 5 materials-15-04040-f005:**
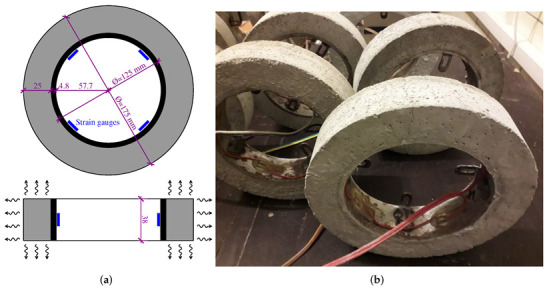
Adopted ring geometry with four strain gauges (**a**) and a cracked ring (**b**).

**Figure 6 materials-15-04040-f006:**
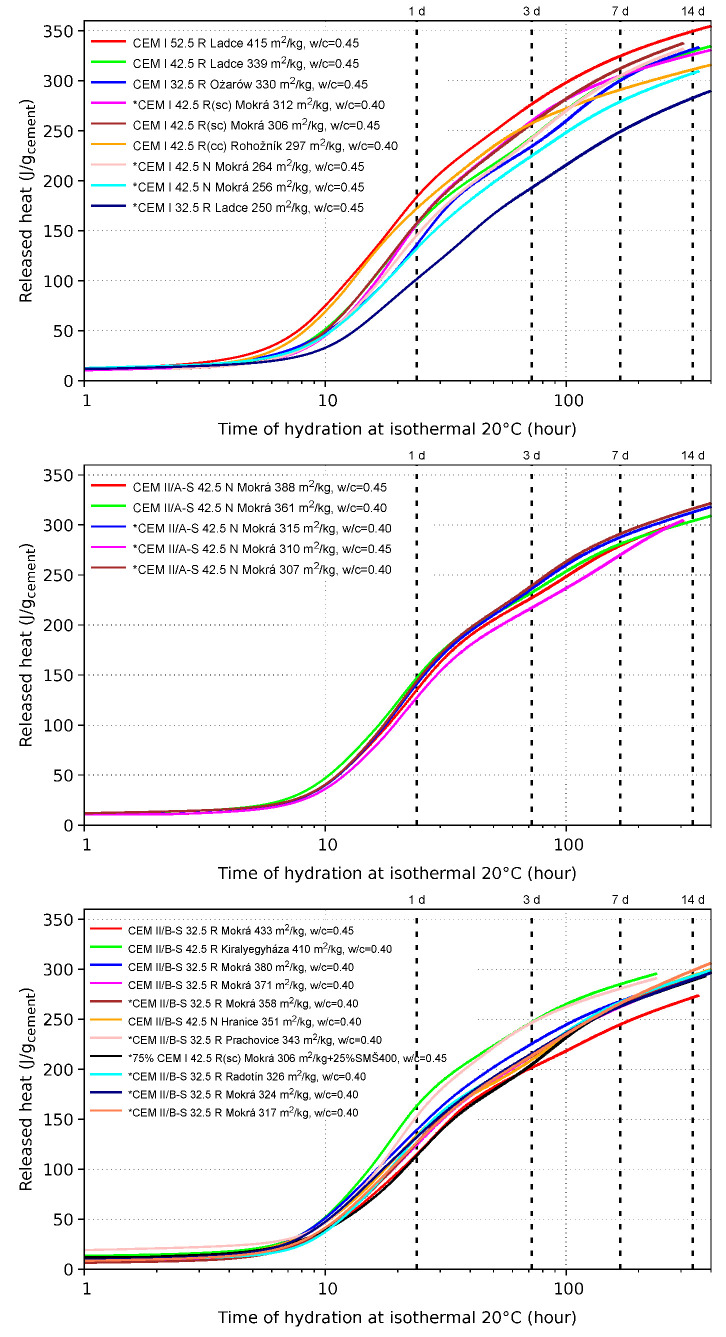
Released heat from isothermal calorimetry on 25 cements from groups CEM I, CEM II/A-S and CEM II/B-S. An asterisk signalizes a crack-resistant cement.

**Figure 7 materials-15-04040-f007:**
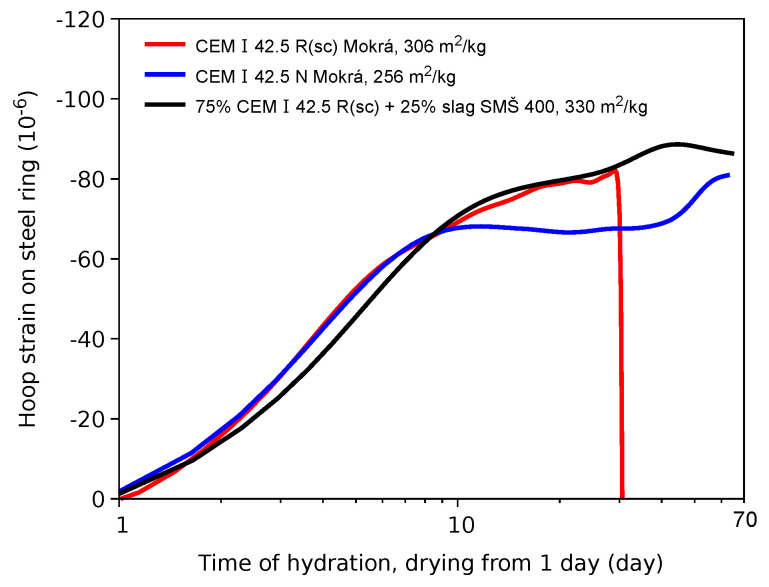
Hoop strain on the steel ring for three selected cements.

**Figure 8 materials-15-04040-f008:**
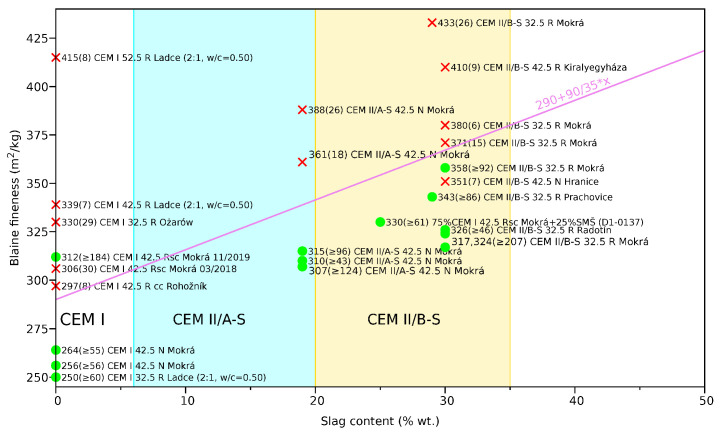
Results from 25 cements measured in the ring test. Each label marks *Fineness(Average_cracking_time) Cement_description*. Green dots mark crack-resistant cements, red crosses mark cements cracking under 40 days in the ring test.

**Figure 9 materials-15-04040-f009:**
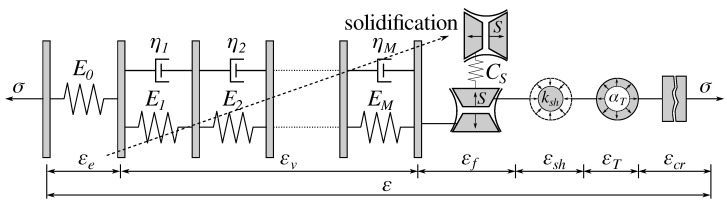
B3 creep model as a solidifying Kelvin chain [39]. Serial coupling with the cracking strain.

**Figure 10 materials-15-04040-f010:**
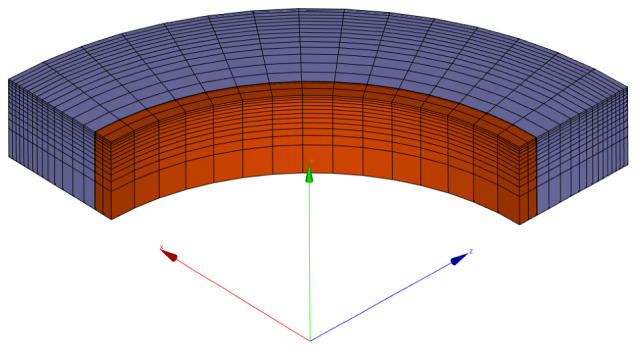
Mesh for hygro-mechanical model showing the restraining steel ring in red and mortar ring in blue.

**Figure 11 materials-15-04040-f011:**
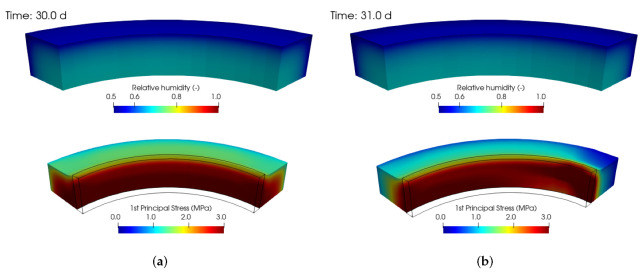
Hygro-mechanical simulation of 1/8 of the ring test. Relative humidity and the first principal stress before cracking (**a**) and after cracking (**b**). Deformations exaggerated 250×.

**Figure 12 materials-15-04040-f012:**
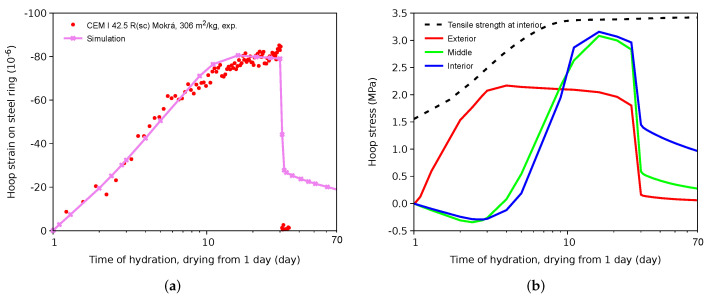
Hoop strain on the steel ring (**a**) and hoop stress on the mid-plane in the vicinity of the artificial shallow notch (**b**).

**Table 1 materials-15-04040-t001:** Chemical composition of selected cements and approximate mineral composition according to Bogue.

Component	CEM I Mokrá	CEM I Ladce	CEM I Prachovice	GGBFS SMŠ 400
SiO_2_	20.7	21.7	18.2	39.7
Al_2_O_3_	4.8	5.3	5.1	6.5
Fe_2_O_3_	3.4	2.8	2.9	0.47
CaO	63.8	66.1	62.8	40.1
MgO	1.4	1.4	2.2	9.5
SO_3_	3.2	3.2	3.2	0.72
K_2_O	0.75	1.0	0.76	0.55
Na_2_O	0.16	0.04	0.31	0.33
Na_2_O-eq.	0.65	0.7	0.82	0.69
LOI	1.46	0.85	4.95	1.25
Insoluble	0.45	-	0.30	-
C_3_S	63.2	59.4	68.5	-
C_2_S	11.7	17.4	0.6	-
C_3_A	7.0	9.3	8.5	-
C_4_AF	10.3	8.6	5.5	-

**Table 2 materials-15-04040-t002:** Relevant properties of tested cements sorted according to the Blaine fineness and cement class.

Cement	Blaine Fineness	Compressive Strength		Released Heat at 24 h	Ring Crack Time ± st. dev.
	(m^2^/kg)	2 d (MPa)	28 d (MPa)	(J/g)	(day)
*CEM I 32.5 R Ladce	250	11.9	37.9	102	≥60±0
*CEM I 42.5 N Mokrá	256	21.1	47.7	133	≥56±0
*CEM I 42.5 N Mokrá	264	21.2	53.3	146	≥55±0
CEM I 42.5 R(cc) Rohožník ^*b*^	297	28.0	56.0	172	8±4
CEM I 42.5 R(sc) Mokrá, March 2018 ^*b*^	306	27.5	59.5	157	30±0
*CEM I 42.5 R(sc) Mokrá, November 2019	312	27.7	59.7	156	≥184±35
CEM I 32.5 R Ożarów ^*b*^	330	21.0	45.0	137	29±5
CEM I 42.5 R Ladce	339	27.1	52.2	156	7±1
CEM I 52.5 R Ladce	415	32.6	58.0	184	8±2
*CEM II/A-S 42.5 N Mokrá	307	22.5	53.1	142	≥124±0
*CEM II/A-S 42.5 N Mokrá	310	18.2	50.4	127	≥43±19
*CEM II/A-S 42.5 N Mokrá	315	21.2	52.4	140	≥96±0
CEM II/A-S 42.5 N Mokrá	361	22.6	52.5	147	18±2
CEM II/A-S 42.5 N Mokrá ^*b*^	388	21.0	54.0	135	26±1
*CEM II/B-S 32.5 R Mokrá	317	14.9	46.6	126	≥207±0
*CEM II/B-S 32.5 R Mokrá	324	16.4	48.5	133	≥207±0
*CEM II/B-S 32.5 R Radotín ^*b*^	326	18	48	133	≥46 ^*a*^
*75% CEM I 42.5 R(sc) Mokrá + 25% SMŠ 400	330	18.0	48.5	113	≥61±5
*CEM II/B-S 32.5 R Prachovice ^*b*^	343	20.1	50.8	154	≥86±6
CEM II/B-S 42.5 N Hranice ^*b*^	351	∼21	∼52	134	7±2
*CEM II/B-S 32.5 R Mokrá	358	19.2	52.3	126	≥92±16
CEM II/B-S 32.5 R Mokrá	371	18.1	53.0	125	15±3
CEM II/B-S 32.5 R Mokrá	380	20.0	53.9	139	6±2
CEM II/B-S 42.5 R Kiralyegyháza ^*b*^	410	18	50	164	9±7
CEM II/B-S 32.5 R Mokrá ^*b*^	433	17	51	115	26±7

^*a*^ The first ring remained uncracked for 77 days. The second one cracked in strange softening steps by 15 days, which is interpreted as a manufacturing defect. An asterisk denotes a crack-resistant cement, ^*b*^ marks a commercially produced cement.

## Data Availability

Data are visualized in figures and provided in tables.

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
