# Peer review of "Crack-Resistant Cements under Drying: Results from Ring Shrinkage Tests and Multi-Physical Modeling"

_materials, 2022, doi:10.3390/ma15124040_

Round 1

Reviewer 1 Report

My comments for the manuscript titled: “Crack-resistant cements under drying: results from ring shrinkage tests and multi-physical modeling”

The performed work is of great importance. It also suits the journal scope. I have some minor comments to improve the manuscript before its publication,

-This type of numbering is not popular, 3000・10−6. It is better to write it as 3000 x 10−6.

-The methodology and experimental process shall be explained further.

-English is poor in some parts, and it is not easy to follow some texts.

-The FEM process done in the software shall be given in more detail. There is no information about boundary conditions, loading, and mech algorithm. It is difficult for future readers to compare their results with those obtained in this paper. Thus, FEM should be detailed.

-How the humidity was distributed through the FEM model should be described.

-Conclusion shall be expanded by giving more details. Some future readers read the conclusion only.

Author Response

Dear reviewer,

we appreciate your time and effort with valuable comments. They will definitely improve readability and expectation of readers. The changes have been marked red in the manuscript. Five reviewers raised several comments and the manuscript went through many changes. The final text was corrected by a native English speaker from the Berlitz Language Center in Prague.

Reviewer #1:

My comments for the manuscript titled: “Crack-resistant cements under drying: results from ring shrinkage tests and multi-physical modeling”. The performed work is of great importance. It also suits the journal scope. I have some minor comments to improve the manuscript before its publication

- This type of numbering is not popular, 3000・10−6. It is better to write it as 3000 x 10−6.

Changed to your suggestion (\times in LaTeX instead of \cdot). Originally, I followed preferences from Germany where × can be confused with x and therefore ・ is preferred. I kept dot products of vectors, this is well established. Fixed on 6 places in the manuscript.

- The methodology and experimental process shall be explained further.

I added Figure 4 for the experimental workflow and modified the text around.

- English is poor in some parts, and it is not easy to follow some texts.

The whole manuscript was corrected and edited by a native English speaker, several paragraphs were rewritten.

- The FEM process done in the software shall be given in more detail. There is no information about boundary conditions, loading, and mech algorithm. It is difficult for future readers to compare their results with those obtained in this paper. Thus, FEM should be detailed.

I added and rephrased several sentences explaining all boundary condition for hygral and mechanical problems. A new figure with the mesh was added, being more obvious than text description.

- How the humidity was distributed through the FEM model should be described.

As now in the text, the initial relative humidity is 0.98 everywhere in the mortar. The Newton boundary condition makes it to converge to 0.50 by surface drying. The relative humidity field is depicted in Figure 10 for 30 and 31 days, showing 2D axisymmetric transport. I believe that it makes no sense to display moisture field evolution in detail since there is no validation. The moisture field is predicted by diffusion equation (2).

-Conclusion shall be expanded by giving more details. Some future readers read the conclusion only.

Conclusion section was completely rewritten, this issue was raised also by other reviewers.

Reviewer 2 Report

In this paper, ring shrinkage tests were carried out on 10 commercial cements and 15 cements produced on demand, and the effects of fineness and slag substitution were evaluated. The conclusions could be referenced by the contractor in cement selection. Generally, the paper was well written. There are the following suggestions:

1.     The introduction section is too long and does not point out the shortcomings of the current research.

2.     The sentence in lines 59-60 needs to be revised.

3.     In the title of Figure 3, is a number missed before the symbol “~”?

Author Response

Dear reviewer,

we appreciate your time and effort with valuable comments. They will definitely improve readability and expectation of readers. The changes have been marked red in the manuscript. Five reviewers raised several comments and the manuscript went through many changes. The final text was corrected by a native English speaker from the Berlitz Language Center in Prague.

Reviewer #2:

In this paper, ring shrinkage tests were carried out on 10 commercial cements and 15 cements produced on demand, and the effects of fineness and slag substitution were evaluated. The conclusions could be referenced by the contractor in cement selection. Generally, the paper was well written. There are the following suggestions:

1. The introduction section is too long and does not point out the shortcomings of the current research.

Introduction section was separated into subsections, some paragraphs left out. Other reviewers suggested to put objectives and new findings, which are there now. The shortcomings were included.

2. The sentence in lines 59-60 needs to be revised.

Those sentences were revised and abridged.

3. In the title of Figure 3, is a number missed before the symbol “~”?

It was an approximate age ~17 years, I rechecked in detail and it was 18 years so the symbol is gone now.

Reviewer 3 Report

Review Comments:

  1. This paper discusses critical aspects of crack-resistant cements under drying.
  2. In abstract it is mentioned that slowly hydrating cements benefit from higher creep, higher stress relaxation and slower chemical shrinkage, all contributing to crack resistant concrete with generally higher durability. The important findings of the research work reported need to be quantitatively expressed rather than in qualitative terms.
  3. Authors need to clearly state the research gaps identified in the work from the summary of the literature survey carried out. Accordingly, novelty of the work carried out along with the objectives of the work need to be clearly stated in the work.
  4. In the Materials and methods section authors shall include the number of specimens prepared for each of the composition of the material considered for the study.
  5. In the Results and discussion, the results obtained (on Isothermal calorimetry, Ring shrinkage test, Compressive strength) from the work need to be appropriately co-related to the already published literature with further elaboration of the behavior of the same.
  6. For the experimental works carried out uncertainty error analysis has to be discussed for the validation of the simulation results presented.
  7. Conclusions section can be highlighted with important findings of the work reported.
  8. The manuscript needs careful corrections for the English grammar and typo errors.

Author Response

Dear reviewer,

we appreciate your time and effort with valuable comments. They will definitely improve readability and expectation of readers. The changes have been marked red in the manuscript. Five reviewers raised several comments and the manuscript went through many changes. The final text was corrected by a native English speaker from the Berlitz Language Center in Prague.

Reviewer #3:

This paper discusses critical aspects of crack-resistant cements under drying.

  1. In abstract it is mentioned that slowly hydrating cements benefit from higher creep, higher stress relaxation and slower chemical shrinkage, all contributing to crack resistant concrete with generally higher durability. The important findings of the research work reported need to be quantitatively expressed rather than in qualitative terms.
    Added explanation to “Hygro-mechanical Model”, demonstrating that increasing 28 day strength (as B3 input parameter of the creep model) leads to decreased both basic and drying creep. This comes from an extensive calibration of Northwestern University database results (over 400 creep tests on different concretes at that time). Since relaxation operator and creep operators are inverse, increasing creep compliance leads to higher stress relaxation. It would be possible to pick up specific tests from Northwestern University database for creep or demonstrate B3 creep prediction on several our mortars. Unfortunately, we did not measure creep of mortars and the model used only default values from B3 model. Chemical shrinkage is manifested by calorimetry results, slower hydration generally means lower chemical shrinkage. Such sentence was added to “Isothermal calorimetry” section.

  1. Authors need to clearly state the research gaps identified in the work from the summary of the literature survey carried out. Accordingly, novelty of the work carried out along with the objectives of the work need to be clearly stated in the work.

I reworked the last paragraph in the introduction, showing objectives, novelty and the gap to close.

  1. In the Materials and methods section authors shall include the number of specimens prepared for each of the composition of the material considered for the study.

I added number of specimens for calorimetry, strength and ring tests. Added workflow of experiments as Figure 4.

  1. In the Results and discussion, the results obtained (on Isothermal calorimetry, Ring shrinkage test, Compressive strength) from the work need to be appropriately co-related to the already published literature with further elaboration of the behavior of the same.

General comparison was added to each section, the cements differ across the literature. Our results are consistent with known trends, such as increasing released heat and strength gain with increased Blaine fineness etc.

  1. For the experimental works carried out uncertainty error analysis has to be discussed for the validation of the simulation results presented.

I added standard deviation of cracking time to Table 2. I have only mean strength values for cements, tested at concrete plants, without the deviation. Both rings for validation, made from CEM I 42.5 R(sc) cement, cracked at 30 days, such a sentence was added.

  1. Conclusions section can be highlighted with important findings of the work reported.

    The conclusion section was completely reworked.

  1. The manuscript needs careful corrections for the English grammar and typo errors.

The whole manuscript was corrected and edited by a native English speaker.

Reviewer 4 Report

Dear Šmilauer et al.,

The manuscript “Crack-resistant cements under drying: results from ring shrinkage tests and multi-physical modeling” (materials-1742709) by Šmilauer et al. show that the restrained ring cracking time generally increases with lower fineness and higher slag substitution. The topic is interesting, but I think this article should reconsider after proper changes in major revision for publication in Materials. Some of my specific comments are below:

  1. In the abstract section (line 1-10), the authors should add quantitative results rather than only qualitative results.
  2. Describe the novelty of the article made by the author? From the results of my evaluation, it seems that many similar published works adequately explain what you have raised in the current manuscript. If there are something others really new in this manuscript, please highlight it more clearly in the introduction section (line 13-144).
  3. The state of the art and the significance of the current study are not clearly present, the authors should highlight it more advanced in the introduction section (line 13-144).
  4. Since this manuscript cement modeling that can be used as fixation in medical implant, I would encourage and advise the authors to adopt some of the specific additional references related to medical implant modeling published by MDPI in the introduction section (line 13-144) as follow:
    • Tresca Stress Simulation of Metal-on-Metal Total Hip Arthroplasty during Normal Walking Activity. Materials (Basel). 2021, 14, 7554. https://doi.org/10.3390/ma14247554
  1. In the materials and methods section (line 115-160), the authors should add one systematic figure to illustrate the workflow of experimental testing in the present study to make the reader more interested and easier to understand rather than only using dominant text to explain.
  2. It is crucial to explain more clearly why ring shrinkage tests needs to be performed.
  3. The author must provide a detailed specification and use condition more detail regarding all tools used in the research carried out so that the reader can estimate the accuracy and differences in the results that the authors describe due to the use of different tools in future studies.
  4. In the Results and discussion section (line 161-258), the authors are advised to compare the results they obtain with previous similar/identical studies if it is possible.
  5. In the last paragraph before conclusion section (after line 258), the authors should add of one paragraph about the limitations of the presented study.
  6. The conclusion (line 259-285) of the present manuscript is not solid. Further elaboration is needed. Also, make it intho paragraph, not point-by-point as in present form.
  7. Further research needs to be explained in the conclusion section (line 259-285).
  8. I see some errors on English in some areas of the present manuscript. To improve the quality of English used in this manuscript and make sure English language, grammar, punctuation, spelling, and overall style are correct, further proofreading is needed. As an alternative, the authors can use the MDPI English proofreading service for this issue.
  9. Please make sure the authors have used the Materials, MDPI format correctly. The authors can download published manuscripts by Materials, MDPI, and compare them with the present author's manuscript to ensure typesetting is appropriate. For example Uppercase and lowercase for the title and all of the section and subsection is not correct.

I am pleased to have been able to review the author's present manuscript. Hopefully, the author can revise the current manuscript as well as possible so that it becomes even better. Good luck for the author's work and effort.

Best regards,

The Reviewer

Author Response

Dear reviewer,

we appreciate your time and effort with valuable comments. They will definitely improve readability and expectation of readers. The changes have been marked red in the manuscript. Five reviewers raised several comments and the manuscript went through many changes. The final text was corrected by a native English speaker from the Berlitz Language Center in Prague.

Reviewer #4:

The manuscript “Crack-resistant cements under drying: results from ring shrinkage tests and multi-physical modeling” (materials-1742709) by Šmilauer et al. show that the restrained ring cracking time generally increases with lower fineness and higher slag substitution. The topic is interesting, but I think this article should reconsider after proper changes in major revision for publication in Materials. Some of my specific comments are below:

1. In the abstract section (line 1-10), the authors should add quantitative results rather than only qualitative results.

I added quantitative results to the abstract.

2. Describe the novelty of the article made by the author? From the results of my evaluation, it seems that many similar published works adequately explain what you have raised in the current manuscript. If there are something others really new in this manuscript, please highlight it more clearly in the introduction section (line 13-144).

This was done in the last paragraph in the Introduction section.

3. The state of the art and the significance of the current study are not clearly present, the authors should highlight it more advanced in the introduction section (line 13-144).

Introduction section was separated into subsections, some paragraphs left out and reworked. Other reviewers suggested to put objectives and new findings which were added.

4. Since this manuscript cement modeling that can be used as fixation in medical implant, I would encourage and advise the authors to adopt some of the specific additional references related to medical implant modeling published by MDPI in the introduction section (line 13-144) as follow:

    • Tresca Stress Simulation of Metal-on-Metal Total Hip Arthroplasty during Normal Walking Activity. Materials (Basel). 2021, 14, 7554. https://doi.org/10.3390/ma14247554

I do not see too much common behavior with implants. Such cements are placed within perfect water-saturated conditions, they actually slightly swell than shrink. Since the driving strain is mainly drying shrinkage, the behavior is different and from our pilot unpublished tests, none of such saturated rings ever broke. I understand it would increase impact of Materials, however, I have not found anything really relevant to the topic and objectives in the archives. The closest match is likely https://doi.org/10.3390/ma14247865 for different concrete curing regimes.

5. In the materials and methods section (line 115-160), the authors should add one systematic figure to illustrate the workflow of experimental testing in the present study to make the reader more interested and easier to understand rather than only using dominant text to explain.

Brilliant idea, I added Figure 4 for the experimental workflow.

6. It is crucial to explain more clearly why ring shrinkage tests needs to be performed.

One sentence was reformulated in the abstract and many paragraphs in the introduction added to emphasize correlation between a short-term ring shrinkage test and long-term concrete performance.

7. The author must provide a detailed specification and use condition more detail regarding all tools used in the research carried out so that the reader can estimate the accuracy and differences in the results that the authors describe due to the use of different tools in future studies.

I added strain gauges and data logger specifications. Table 2 was supplemented with standard deviation for cracking time. The experiments could be replicated now without any difficulty, I see no fundamental missing information now.

8. In the Results and discussion section (line 161-258), the authors are advised to compare the results they obtain with previous similar/identical studies if it is possible.

A few sentences were added to isothermal calorimetry subsection and one more paragraph to the ring tests.

9. In the last paragraph before conclusion section (after line 258), the authors should add of one paragraph about the limitations of the presented study.

General limitations were listed already in subsection “Microtomography and ring tests” since it was raised by another reviewer. I added one paragraph to the hygro-mechanical model, the major issue are missing parameters which would need experimental calibration.

10. The conclusion (line 259-285) of the present manuscript is not solid. Further elaboration is needed. Also, make it into paragraph, not point-by-point as in present form.

The conclusion section was completely reworked.

11. Further research needs to be explained in the conclusion section (line 259-285).

A sentence was added to the conclusion section “Further research may include detailed experimental identification of material properties, microcrack characterization and crack resistance of blended cements.”

12. I see some errors on English in some areas of the present manuscript. To improve the quality of English used in this manuscript and make sure English language, grammar, punctuation, spelling, and overall style are correct, further proofreading is needed. As an alternative, the authors can use the MDPI English proofreading service for this issue.

The whole manuscript was corrected and edited by a native English speaker.

13. Please make sure the authors have used the Materials, MDPI format correctly. The authors can download published manuscripts by Materials, MDPI, and compare them with the present author's manuscript to ensure typesetting is appropriate. For example Uppercase and lowercase for the title and all of the section and subsection is not correct.

Thank you for pointing that out, changed according to your comments. Also, the figures side-by-side were marked (a) and (b) instead of left and right, which seems to be more common in Materials.

I am pleased to have been able to review the author's present manuscript. Hopefully, the author can revise the current manuscript as well as possible so that it becomes even better. Good luck for the author's work and effort.

Reviewer 5 Report

The paper is very well written is easy to read. I commend the authors for an elaborate and exhaustive work carried out. The results have been presented and explained well and is of high importance to the readers of the journal. 

That being said, the reviewer feels that there are some grammatical errors in the manuscript and advise the authors to correct them for better readability. 

Comments from reviewer:

1.     Page 1, Line 18: “Experiments on cm-sized samples led to linear shrinkage strains over 3000x10−6, 1500x10−6and 500x10−6 at 500 days for the cement paste, mortar 19 and concrete, respectively” Is there a numerical value specifying the size of sample, missing? 

2.     Figure 1: Does the plot pertain to average compressive strengths of 193 and (>2150) cements produced 1950’s and 1994, respectively. If so, kindly elaborate and specify how the average have been taken if there are different exposure conditions?

3.     Page 2, Line 47: “Based on that research, three simple tests were proposed for crack-resistant cement; concrete compressive strength below 7 MPa at 12 hours of semi-adiabatic curing, chemical shrinkage below 1.05 ml per 100 g of cement at 12 hours and visual inspection of dried cement pastes” The reviewer believes they are not simple tests but general observations. Also, they were proposed to identify crack-resistant cement.

4.     Page 5, Line 144: “14 pastes were tested correctly at w/c=0.40, 144 11 pastes with a higher w/c=0.45 due to compatibility with the mortar in the rings” Kindly elaborate the compatibility requirements.

5.     Kindly explain a sample notation used in the Figure 7. For example, which value in the labels indicate the average cracking time.

6.     A lot of expressions used in the manuscript have not been defined in the parenthesis. Kindly ensure they have been properly explained.

7.     What is the basis of equation 8, kindly cite the source

8.     A brief note on the simulations might help the readers understand Figure 9 in detail.

9.     Is there a specific reason why authors chose to display hoop strain and hoop stress plots for CEM1 42.5R(sc) Mokra, 306 m2/kg? This Figure seems out of place. 

Author Response

Dear reviewer,

we appreciate your time and effort with valuable comments. They will definitely improve readability and expectation of readers. The changes have been marked red in the manuscript. Five reviewers raised several comments and the manuscript went through many changes. The final text was corrected by a native English speaker from the Berlitz Language Center in Prague.

The paper is very well written is easy to read. I commend the authors for an elaborate and exhaustive work carried out. The results have been presented and explained well and is of high importance to the readers of the journal. 

That being said, the reviewer feels that there are some grammatical errors in the manuscript and advise the authors to correct them for better readability. 

Comments from reviewer:

1.     Page 1, Line 18: “Experiments on cm-sized samples led to linear shrinkage strains over 3000x10−6, 1500x10−6and 500x10−6 at 500 days for the cement paste, mortar 19 and concrete, respectively” Is there a numerical value specifying the size of sample, missing? 

I put there specific dimensions of the prisms.

2.     Figure 1: Does the plot pertain to average compressive strengths of 193 and (>2150) cements produced 1950’s and 1994, respectively. If so, kindly elaborate and specify how the average have been taken if there are different exposure conditions?

As far as I know, the exposure conditions were the same, curing specimens in a water bath. However, older ASTM C109 used equivalent flow rather than water/cement ratio, so the results differ. I added the sentence explaining that 1-to-28 strength ratio has obviously increased, which is the point.

3.     Page 2, Line 47: “Based on that research, three simple tests were proposed for crack-resistant cement; concrete compressive strength below 7 MPa at 12 hours of semi-adiabatic curing, chemical shrinkage below 1.05 ml per 100 g of cement at 12 hours and visual inspection of dried cement pastes” The reviewer believes they are not simple tests but general observations. Also, they were proposed to identify crack-resistant cement.

Indeed, I replaced ‘simple tests’ with ‘criteria’. ‘Simple test’ came from the title “Three simple tests for selecting low-crack cement”, Cement and Concrete Composites 2004, which can be confusing.

4.     Page 5, Line 144: “14 pastes were tested correctly at w/c=0.40, 144 11 pastes with a higher w/c=0.45 due to compatibility with the mortar in the rings” Kindly elaborate the compatibility requirements.

The sentence was rewritten: 14 pastes were tested according to prEN 196-11 at w/c=0.40, 11 pastes had a higher w/c=0.45 matching mostly the ring's mortars.

5.     Kindly explain a sample notation used in the Figure 7. For example, which value in the labels indicate the average cracking time.

You are right. I put inside the caption: Each label codes Fineness(Average_cracking_time) Cement_description.

6.     A lot of expressions used in the manuscript have not been defined in the parenthesis. Kindly ensure they have been properly explained.

All equations were checked so the terms are defined (beyond obvious stress and strain vectors).

7.     What is the basis of equation 8, kindly cite the source

You probably mean equation 5, the reference was added, the other equations have references as well.

8.     A brief note on the simulations might help the readers understand Figure 9 in detail.

More explanation was provided and some sentences rephrased.

9.     Is there a specific reason why authors chose to display hoop strain and hoop stress plots for CEM1 42.5R(sc) Mokra, 306 m2/kg? This Figure seems out of place.

Hoop strain evolution is the only validated output with the cracking time. This cement is a good representative since it lies on the border between crack resistant and crack prone cement. Also, there is no scatter in the cracking time in both rings. The stress was put there to illustrate quite general evolution influenced by drying, creep and microcracking.

Round 2

Reviewer 4 Report

It is improved, well done.